# Applying Tissue Slice Culture in Cancer Research—Insights from Preclinical Proton Radiotherapy

**DOI:** 10.3390/cancers12061589

**Published:** 2020-06-16

**Authors:** Theresa Suckert, Treewut Rassamegevanon, Johannes Müller, Antje Dietrich, Antonia Graja, Michael Reiche, Steffen Löck, Mechthild Krause, Elke Beyreuther, Cläre von Neubeck

**Affiliations:** 1German Cancer Consortium (DKTK), Partner Site Dresden and German Cancer Research Center (DKFZ), 69120 Heidelberg, Germany; Treewut.Rassamegevanon@uniklinikum-dresden.de (T.R.); Antje.Dietrich@uniklinikum-dresden.de (A.D.); Antonia.Graja@uniklinikum-dresden.de (A.G.); steffen.loeck@oncoray.de (S.L.); Mechthild.Krause@uniklinikum-dresden.de (M.K.); Claere.VonNeubeck@uk-essen.de (C.v.N.); 2OncoRay—National Center for Radiation Research in Oncology, Faculty of Medicine and University Hospital Carl Gustav Carus, Technische Universität Dresden, Helmholtz-Zentrum Dresden—Rossendorf, 01309 Dresden, Germany; Johannes.Mueller@uniklinikum-dresden.de (J.M.); Michael.Reiche@uniklinikum-dresden.de (M.R.); Elke.Beyreuther@uniklinikum-dresden.de (E.B.); 3Institute of Radiooncology—OncoRay, Helmholtz-Zentrum Dresden—Rossendorf, 01328 Dresden, Germany; 4National Center for Tumor Diseases (NCT), Partner Site Dresden, 01307 Dresden, Germany; 5Department of Radiotherapy and Radiation Oncology, Faculty of Medicine and University Hospital Carl Gustav Carus, Technische Universität Dresden, 01309 Dresden, Germany; 6Helmholtz-Zentrum Dresden—Rossendorf, Institute of Radiation Physics, 01328 Dresden, Germany; 7Department of Particle Therapy, University Hospital Essen, University of Duisburg-Essen, 45147 Essen, Germany

**Keywords:** tumor biology, thin-cut tissue slices, proton beam radiotherapy, head and neck cancer, organotypic brain slice culture, DNA damage

## Abstract

A challenge in cancer research is the definition of reproducible, reliable, and practical models, which reflect the effects of complex treatment modalities and the heterogeneous response of patients. Proton beam radiotherapy (PBRT), relative to conventional photon-based radiotherapy, offers the potential for iso-effective tumor control, while protecting the normal tissue surrounding the tumor. However, the effects of PBRT on the tumor microenvironment and the interplay with newly developed chemo- and immunotherapeutic approaches are still open for investigation. This work evaluated thin-cut tumor slice cultures (TSC) of head and neck cancer and organotypic brain slice cultures (OBSC) of adult mice brain, regarding their relevance for translational radiooncology research. TSC and OBSC were treated with PBRT and investigated for cell survival with a lactate dehydrogenase (LDH) assay, DNA repair via the DNA double strand break marker γH2AX, as well as histology with regards to morphology. Adult OBSC failed to be an appropriate model for radiobiological research questions. However, histological analysis of TSC showed DNA damage and tumor morphological results, comparable to known in vivo and in vitro data, making them a promising model to study novel treatment approaches in patient-derived xenografts or primary tumor material.

## 1. Introduction

Radiotherapy is an integral part of anti-tumor therapy in >50% of the patients [1]. Proton beam radiotherapy (PBRT) has evolved as a therapy option that is particularly relevant for pediatric cancer patients and tumors that are surrounded by critical normal tissue structures, e.g., head and neck cancers affecting the brain stem, optical nerve, parotid gland, and swallowing muscles [2,3]. Attributed to the depth dose profile of the proton beam, the critical normal tissue behind the tumor receives very low to no dose, while the tumor can be covered with iso-effective doses relative to conventional radiotherapy with photons [4]. A potential benefit of PBRT is therefore the reduced toxicity in the normal tissue.

Although the number of PBRT centers are increasing worldwide [5], additional experimental data, e.g., on combination with drugs (chemotherapy, small molecules, inhibitors), tumor microenvironment, and alternative fractionation schedules, are needed to explore the full potential of this therapy [6,7,8]. Like cancer research in general, PBRT faces the problem of defining reproducible, reliable, and practical models of tumors and normal tissues, which reflect the effects of complex treatment modalities and the heterogeneous response of patients [9]. Less extensive alternatives to standard in vivo models for organ toxicity and tumor response, which also exhibit a low intra-model variation even in small sample sizes, need to be identified.

Squamous cell carcinoma of the head and neck (HNSCC) constitutes more than 90% of all head and neck cancers arising from the oral cavity, larynx, nasopharynx, oropharynx, hypopharynx, and salivary glands. HNSCC affects 4.9% of all cancer patients worldwide, and accounts for 4.8% of cancer related deaths [10]. The high mortality rate underlines the medical need for more effective treatment solutions. The dosimetric advantage of intensity modulated PBRT relative to photon-based radiotherapy has been recognized [3], and is under investigation regarding toxicity reduction in the normal tissue, allowing for potential dose escalation to these tumors. Patient-derived and cell line-derived HNSCC models were shown to recapitulate the parental tumor characteristics, and are suitable for chemotherapy and radiotherapy testing [11,12,13,14]. However, the immunocompromised mice hosting the tumor have high demands, and request individual solutions for radiation exposure [15]. In addition, animal experiments are time-extensive, and experimental beamtime is usually limited at proton facilities.

One model to bridge the gap between in vivo and conventional in vitro models is thin-cut slices, such as tumor slice culture (TSC), which preserve the original three-dimensional (3D) structure of the organ and closely mimic the in vivo situation. Ex vivo cultures of neonatal or postnatal (1 week) organotypic brain slice culture (OBSC) have been used for decades, and are a well-accepted model for the field of developmental biology [16]. Nevertheless, their transferability to adult tissue is debatable, leading to an increased effort in the cultivation of brain slice culture from adult mice [17,18,19]. In the context of translational cancer research, OBSC have been used to study tumor invasion and micromilieu [20,21]; possible applications in radiooncology are, e.g., to examine the much-debated effects of PBRT on migratory behavior [22,23,24].

This work evaluated thin-cut tumor slice cultures of a HNSCC xenograft [13] and organotypic slice cultures of adult mouse brain regarding their relevance for translational research questions in PBRT. As functional endpoints, cellular survival, metabolic activity, and morphology of different cell populations were analyzed, following proton beam irradiation. The DNA double strand break marker γH2AX was used to quantify the radiation response.

## 2. Results

### 2.1. Optimized Angle for Tissue Slice Irradiation

The experimental area of the University Proton Therapy Dresden (UPTD) is equipped with a horizontal beam line, making it necessary to irradiate the samples in an angular position. To ensure the homogenous radiation exposure of the tissue slices, we established a suitable angle with a rapid prototype constructed with Lego^®^ bricks. Stability against tilting and sufficient medium coverage during treatment was tested for at least 5 min for angles between 42°–90° relative to the beam axis (Figure 1a and Appendix A). The evaluation of irradiated EBT3 films revealed that the dose within the individual membrane inserts was homogeneous, irrespective of the medium volume (1.0–1.5 mL) within the inserts (Appendix A). For achieving dose homogeneity of ≥95% across the whole plate, a horizontal plate direction (Figure 1a) was superior to a vertical plate direction (Appendix A), when angles of ≥70° were applied. Optimal dose homogeneity would be reached by upright irradiation. Considering the other parameters, however, an angle of 76° (Figure 1b) was finally chosen for the proton irradiation. After validating the thin-cut tissue slice models, a milled, more rigid version of the setup was constructed (Figure 1c and Appendix A). A mechanical drawing (Appendix A) and blueprints (Appendix A) can be found in the supplement.

### 2.2. Tumor Slice Culture

Vibratome cutting conditions were optimized for the used Cal33 xenograft tumors grown on the hindleg of NMRI mice (for parameters see Section 4.3.). Cell viability measured by a resazurin based assay (PrestoBlue™) showed metabolic activity of the TSC over six days (Figure 2); during this period, the slice shape remained unchanged, and no deterioration of the extracellular matrix was observed (data not shown). The histological tissue morphology after two days ex vivo corresponded to the in vivo situation: slices maintained expression of the cancer stem cell (CSC) marker CD44 (Figure 3a), a necrotic and hypoxic center (Figure 3a,b), as well as an oxic rim with proliferating, BrdU positive cells (Figure 3c). Proliferation over the whole six-day culture period was validated by Ki67 staining (Figure 3d). In order to capture in vivo conditions as closely as possible, and compare the results to the existing literature data, the irradiation time point was defined as 24 h after sample preparation.

TSC were analyzed for biomarker changes following irradiation and along their physiological depth from skin to leg muscle. Neither the staining of total cells (DAPI), necrosis (H&E), or the CSC marker CD44 revealed significant differences between irradiated and control slices at 24 h post irradiation. However, independent of treatment, TSC maintained a typical tumor heterogeneity across the physiological tissue depth (Figure 4a). The necrotic fraction of tumor slices (49.5 ± 7.9%) correlated negatively with depth in the tissue (Pearson correlation R = −0.88, *p* < 0.01), CD44 expression (R = −0.61, *p* = 0.02) and the DAPI-positive area (R = −0.71, *p* < 0.01). Significant positive correlations were found between the CD44-positive staining and DAPI-positive area (R = 0.76, *p* < 0.01), tissue depth and CD44 (R = 0.67, *p* < 0.01), and depth and DAPI-positive area (R = 0.83, *p* < 0.01, n = 15, Figure 4b).

Cytotoxicity was measured via lactate dehydrogenase (LDH) release into the supernatant for two weeks after exposure on day 4 with 0 Gy, 10 Gy, or 20 Gy. For the three groups, high LDH levels indicating increased cell lysis were measured during the whole cultivation period. There was no detectable difference between irradiated and control samples (Appendix A).

HNSCC cells stained for the DNA double strand break marker γH2AX showed definite foci, few apoptotic cells, and rare mitotic events (Figure 5a). The applied automated foci detection method was confirmed against manual counting by two independent observers (TR, TS). Bland-Altman plots were used to compare inter-observer differences (Appendix A), and the difference of manual counting and the algorithm (Appendix A). Both groups had a bias close to zero, and a standard deviation of five. Observers deviated most at very high foci numbers (>20), whereas the algorithm tended to miscount cells with very low or high foci numbers. While the manual counting yielded very precise foci numbers, the algorithm evaluation occurred at a considerably higher speed and enabled high-throughput analysis.

Irradiated TSC showed significantly enlarged HNSCC cell nucleus areas (Figure 5b), suggesting a radiation-induced cell cycle arrest. No difference between nucleus areas among slices of the treated and untreated group was found (Appendix A). The number of γH2AX foci was consequently normalized to the ratio of the mean nucleus area of the respective treatment group and the individual nucleus area (cfoci). There was a significant increase in cfoci after proton irradiation with 4 Gy (*p* < 0.01), while an insignificant heterogeneity in the cfoci was found across the depth of the tumor (Appendix A).

### 2.3. Organotypic Brain Slice Culture

The evaluation of the inflammatory cytokine IL6 in the supernatant of OBSC revealed a strong initial inflammatory reaction (mean with SD = 1769 ± 203 pg/mL), which decreased at day 4 in culture (Appendix A); thus, this time point was determined for irradiation experiments.

OBSC were irradiated with the described setup with doses ranging from 10–35 Gy. The cellular cytotoxicity determined by the LDH release into the supernatant showed no increase in cell death at any time point or irradiation dose (Appendix A). However, a concentration-dependent cell death could be induced with Triton X, showing the general functionality of the assay. None of the analyzed cytokines (IL1α, IL1β, IL6, IL10, IL12p70, IL17A, IL23, IL27, MCP1, IFNβ, IFNγ, TNFα, and GM-CSF) were found to be increased after irradiation or lipopolysaccharide stimulation (data not shown).

The composition of neurons (NeuN) and myelin (OSP) remained stable over five days in culture. However, the cell morphology changed dramatically, showing shrinking nuclei and reduced dendrite density (MAP2) at day 5 (Figure 6a–c). A global cell death of microglia (Iba1), as well as atypical astrocytes (GFAP) were observed, irrespective of radiation (Figure 6d–f). Remaining microglia were mainly in amoeboid shape, indicating a strong activation. Astrocytes along blood vessels and outer brain tissue had reduced extensions, hence, losing their star-shaped appearance (Figure 6e). The overall astrocyte density decreased, however, some of them had an excess proliferation, leading to newly formed, unnatural cell clusters (Figure 6e). Radiation exposure up to 35 Gy did not alter the morphology and cell composition of OBSC.

Gamma H2AX foci formation following radiation revealed that OBSC responded to DNA damage, proving that cells were still metabolically active (Figure 7a). However, the typical spot-like formation of foci were largely absent, and foci evaluation could only be performed in a binary manner, i.e., cell nuclei positive or negative for γH2AX foci. Endogenous γH2AX expression was present in unexposed control slices, but proton irradiation caused a significant increase in the number of γH2AX positive cells (Figure 7b).

## 3. Discussion

Preclinical cancer research relies on in vitro, in vivo, and ex vivo models, and their translational potential for the clinic. Thin-cut tissue slices have the advantage of a relatively fast and inexpensive cultivation with straightforward culture procedures, and the possibility to investigate cell interactions within the original 3D architecture [25]. Furthermore, it provides the opportunity to greatly reduce animal numbers in experiments [26]. Tumor and brain slice cultures have been successfully applied for drug screening [25,27,28,29,30,31] and radiation experiments [28,30,31], but so far not in the context of PBRT. In this study, we described an experimental setup for proton irradiation of TSC and OBSC, and clinically evaluated the relevant endpoints regarding radiooncological research. Our investigation aimed to reduce the number of animals needed to obtain meaningful results by testing thin-cut slice cultures as models of high translational value for HNSCC tumors, as well as for normal tissue.

### 3.1. Tumor Slice Culture

Tumor slice culture could be a powerful tool for personalized medicine. Primary patient material can be processed immediately as TSC [25,29,30], or may be expanded as patient-derived xenografts, before screening for the optimal treatment. Therapy response and biomarker analysis can then help to stratify the patients into smaller subgroups [32]. We used Cal33 xenografts as a pilot model for HNSCC and investigated residual γH2AX foci and pathomorphological properties 24 h post proton irradiation, as well as cell survival over 14 days.

TSC maintained their viability for at least six days (Figure 2) and recapitulated the in vivo morphology of Cal33 tumors and HNSCC xenograft tumors, showing a central necrosis with a fraction of hypoxic cells surrounded by a proliferating rim of viable tissue [13,33] (Figure 3). The 3D architecture of the tumor tissue remained stable without a sign of disintegration or extracellular matrix breakdown.

Tumor growth delay [34], tumor regression [35], and local tumor control [13] are important experimental endpoints in translational radiation oncology research using xenografts. For conventional 2D and 3D cell culture models, these endpoints can be mimicked with cellular survival in a colony formation assay [36,37] or a spheroid control dose experiment with growth delay and regrowth analysis [38]. The establishment of comparable endpoints for TSC is therefore of high relevance. However, a prerequisite for these endpoints is the usage of specific cell numbers or spheroid sizes as starting criteria to evaluate therapy responses of clonogenic units. Due to the limited cultivation period and the 3D architecture of the TSC, there are no comparable assays available so far, and surrogate tests had to be performed. Although the LDH assay delivered fast results, it is error prone, and not able to capture minor changes in cell survival [39]. Furthermore, a strong influence (i.e., LDH deterioration) of sample storage on the results was observed. The central necrosis of Cal33 tumors (49.5 ± 7.9%, Figure 3b) induced a very high baseline LDH release in TSC (Appendix A), and continuous cell proliferation (Figure 3c,d) further complicated the interpretation of the results.

Heterogeneous tumor composition is well known from clinical and preclinical studies [14,25,29,40,41], and needs to be taken into consideration when the results are interpreted. This attribute was also preserved in TSC: cell density, the CSC marker CD44, and the necrotic fraction were highly dependent on the position and depth within the original tumor, as well as the distance to the formerly supplying vessels (Figure 4). Biometrical planning of experiments needs to consider heterogeneity in tumor tissue for statistically meaningful results. In particular for TSC, consecutive slices should be assigned to different treatment conditions, and the comparability of evaluated tumor regions needs to be ensured [25]. Moreover, higher replicate numbers should be used for robust results. The intra-model variation represents a disadvantage when it comes to limited experimental time at PBRT sites, however, offers a realistic view into a complex biological system. In future experiments, additional radiation doses and tissue markers, e.g., for cell death [30], could be added. Radiation-induced biomarker alterations, such as a decrease in CSCs, may be revealed by choosing later time points than 24 h post irradiation [25], or performing kinetic measurements over several days [27]. This could be tested as a surrogate marker for tumor shrinkage in future experiments. Nevertheless, due to the ongoing proliferation that was proven by CD44 and BrdU staining, this might not be the most optimal endpoint. Alternatively, short time points could be analyzed with higher specificity, and in greater detail with methods such as laser microdissection microscopy, MALDI imaging, or single-cell ‘omics’ analysis.

The γH2AX assay is a gold standard in radiobiology. It is used in all assay systems of the translational chain—2D, 3D, in vivo, in patient samples, in tissues and blood cells—showing quantitative and qualitative comparable results [14,33,41,42,43,44,45,46,47]. Thus, it is even used in radiation countermeasures to calculate the applied radiation dose. In radiation oncology, the γH2AX foci assay assesses the radiation response in preclinical and clinical in vivo and ex vivo tumor samples [14,33,42,44,48] by quantifying the persisting (24 h post irradiation) radiation-induced DNA damage, leading to mitotic catastrophe or senescence. We could successfully apply the assay in TSC (Figure 5) and observed a radiobiological response similar to previously published in vivo, in vitro, and ex vivo γH2AX foci results: radiation caused a significant increase in γH2AX foci numbers in TSC [31,33,49], and enlarged cell nucleus areas point towards cell cycle effects [50]. On the other hand, no statistically relevant differences were observed within the two treatment groups, neither for nucleus area (Appendix A) nor corrected foci (Appendix A), making it a strong predictor irrespective of tumor heterogeneity. The successfully developed semi-automated foci counting algorithm facilitates the counting process, making the γH2AX foci assay more accessible to inexperienced users.

Another model promising high translational value is tumor organoids, which mimic the 3D architecture of the original tumor and serve as an useful tool in cancer research e.g., for chemotherapy testing [51,52]. Nevertheless, similar to 3D spheroid cultures, organoids are—albeit heterogeneous—monocultures and, therefore, are lacking the typical microenvironment created by multi-cellular complex tumors. In radiation research, the crosstalk between tumor and the tumor associated cells is of high importance, which is not reflected by organoids. A direct comparison of radiation induced changes in TSC and organoids was outside the scope of this study, but would be interesting to investigate in future experiments, since promising protocols for the establishment of HNSCC patient derived organoids are arising [53].

For the current study, only a limited amount of animals bearing the right tumor size was available during proton beam time; a restriction common in proton radiobiology experiments. The strength of the presented results is that, despite this restraint, a variety of questions could be addressed; hence, emphasizing the use of TSC for following the 3R principles.

### 3.2. Organotypic Brain Slice Culture

Radiation therapy can induce early and late side effects in the normal tissue surrounding the tumor. The main potential advantage of PBRT is the reduced normal tissue toxicity. Thus, models of normal tissue response are as important as tumor models in this field. While early side effects, such as nausea and headaches [54], are generally reversible, late side effects, like brain necrosis [55] or cognitive decline [56], persist, influencing quality of life and overall health condition. On a cellular level, early and delayed side effects are the break-down of the blood-brain barrier, astrogliosis, and microglia activation [57]. While the blood-brain barrier cannot be modelled with OBSC, astrogliosis and microglia are accessible endpoints. We investigated if OBSC could be used as an appropriate model for the normal tissue response following radiation.

In response to ex vivo cultivation, astrocytes lose their star-shaped morphology and show ambiguous growth behavior, exhibiting both reduced cell density and sporadic, uncontrolled proliferation (Figure 6d,e). The number of microglia decreased even without additional treatment (Figure 6f). Remaining cells display an amoeboid morphology, which is generally associated with a highly activated state [58]. Activated microglia have been shown to promote tumor invasion in astrocytic glioma and glioblastoma [59,60], and lead to normal tissue alterations, such as cell fate changes of hippocampal stem cells [61]. The observed activation or death (Figure 6f), as well as the apparently limited abilities of OBSC to produce inflammatory cytokines, strongly impedes their suitability for investigating tumor invasion and brain tissue toxicities. Morphological and functional changes in adult OBSC during long-term cultivation have been shown before [16,19,62], and represent a major drawback of the model. Others circumvent this problem by performing tumor invasion experiments within four days after slice preparation [20]. However, the rather intense inflammatory reaction within the first days after OBSC initialization (Appendix A) postponed a reasonable experiment start to day 4.

After radiotherapy, radiation-induced cell death appears in cancer patients. Even though the LDH assay has been shown to accurately determine neuronal apoptosis [63], an increase in cell death (LDH release, Appendix A) did not occur in the here presented experiments for single irradiation doses as high as 35 Gy within an observation period of 14 days, either due to the mentioned model restrictions or insufficient assay sensitivity. However, DNA damage repair as an immediate reaction to irradiation could be initiated. Quantification of radiation (10 Gy protons) induced γH2AX signal showed increased numbers of marker positive cells, but induced less distinct DNA damage foci. The typical spot-like appearances of the γH2AX foci signal was largely missing, and individual counting was not possible, revealing yet another limitation of the model.

Alternatives to the adult OBSC used here are brain organoids or slices derived from neonatal mice. Stem-cell derived organoids (e.g., so-called “mini-brains”) promise to revolutionize the field of personalized medicine [64], but few experiments on oncological research questions have been conducted so far [51,65,66], and standardization is needed before moving to routine clinical practice. Thin-cut tissue slices from neonatal mice are easily accessible, and have been successfully used to study tumor microenvironment and migration in brain tissue before [21]. For investigating normal tissue toxicities after radiation, however, their application appears to be problematic. Although proton therapy is the method of choice for pediatric brain tumors, most patients are juveniles or adults with matured brain tissue. The representativeness of neonatal tissues for research questions concerning adult patients is questionable, and it remains open to which extend the observed aging in culture [28] resembles in vivo aging. The translational value of neonatal brains seems therefore lower than adult brain tissues. A comparison of adult and neonatal OBSC has been done before [62], demonstrating limitations in the long-term culture of the mature brain and differences in cytoarchitecture, cellular composition, and metabolism in neonatal tissue. We found that adult brain slices did not show a proper inflammatory reaction or DNA damage repair, and the cell morphology was impaired due to the cultivation. Our study adds to the findings of Staal et al., underlining the questionable use of long-term adult brain slice cultures [62]. For investigating normal tissue complication in adult brain after radiotherapy, we therefore see the necessity for in vivo experiments, which, so far, yield very promising results [67].

## 4. Material and Methods

### 4.1. Animals

The animal facility and the experiments were approved according to the European Parliament and Council (EU Directive 2010/63/EU) on the protection of animals used for scientific purposes, the German animal welfare regulations, and to the local ethics committee (approval brain slice culture: TV 9/2017; tumor slice culture: TVV 11/2015, Dresden, Germany).

For the OBSC, six to eight weeks old male C57BL/6 were obtained from Janvier Labs (Le Genest-Saint-Isle, France). Cryo-conserved tumor pieces of a tongue squamous carcinoma model Cal33 (DMSZ, Braunschweig, Germany) were thawed and transplanted subcutaneously on the hind-leg of NMRI (nu/nu) mice, as described previously [13]. The authenticity of the cell line was confirmed by microsatellite analysis, histology, and volume doubling time. Animals were kept grouped in Euro Standard Type III cages at 12:12 h light-dark cycle with food and water ad libitum. Animals were sacrificed by cervical dislocation prior to organ removal.

### 4.2. Cell Culture Media

Reagents and composition of cell culture media are listed in Table 1.

### 4.3. Thin-Cut Tissue Slice Culture

For TSC, tumors were excised and immediately put on ice-cold DMEM supplemented with 1× antibiotic-antimycotic. After embedding tumors in 3–4% low melting point agarose (R0801, Life Technologies, Darmstadt, Germany) dissolved in DMEM, cutting was performed on the Leica Vibratome VT1200 S (amplitude 3 mm, speed 0.7 mm/s) in ice-cold PBS and 1× antibiotic-antimycotic. Tumor slices of 500 µm thickness were cut from skin to muscle tissue and immersed in ice-cold DMEM complete medium (see Table 1).

Excised brains for OBSC were immediately transferred to ice-cold dissection buffer (see Table 1) and subsequently embedded in 2–4% low melting point agarose, dissolved in dissection buffer without B27 supplement. Axial brain slices of 300 µm thickness were cut (amplitude 1 mm, speed 0.12–0.16 mm/s) from rostral to caudal.

After cutting, TSC and OBSC slices were placed on PTFE cell culture inserts (PICM03050, Merck, Darmstadt, Germany) and transferred to 6-well plates containing DMEM complete or cultivation medium (see Table 1), respectively, with PBS buffer in between the wells, to avoid medium evaporation. Slices were maintained at 37 °C in humidified atmosphere with 5% CO_2_. The respective medium was exchanged 24 h post cutting and every other day thereafter. The retrieved medium was either stored at 4 °C and processed within 24 h or frozen at −80 °C for later analysis.

For experiments with TSC, consecutive slices were allocated to the treatment arm or control arm. Four hours prior to irradiation, the proliferation marker BrdU (5 µM, clone Bu20a, Agilent Technologies, Hamburg, Germany) and the hypoxia marker pimonidazole (8.5 µM, HP3, Hypoxyprobe, Burlington, MA, USA) were added to the culture medium, and replaced by DMEM complete medium 1 h post irradiation.

Under comparative condition, corresponding OBSC brain halves served as sample and internal control. For cell death induction, 0.1% or 1% Triton X was added to the culture medium for 4 h.

### 4.4. Irradiation Experiments

Proton irradiation was performed at the horizontal fixed-beam beam line in the experimental hall of the UPTD. For 150 MeV protons, a dedicated beam shaping system consisting of a double-scattering device and a ridge filter provides a laterally extended 10 × 10 cm^2^ proton field and a spread-out Bragg peak (SOBP) of 26.3 mm (90% dose plateau) in water [68]. The experimental setup, beam parameters and daily quality assurance of beam delivery are described in detail elsewhere [69]. Lateral dose homogeneity of the proton field was assured by 2D dose measurements with the Lynx scintillation detector (IBA Dosimetry, Schwarzenbruck, Germany), whereas proton dose delivery was controlled by a segmented beam transmission chamber (model 34058, PTW, Freiburg, Germany) at beam exit. The latter was calibrated daily against a capped Markus ionization chamber (model 34045, 1.06 mm water equivalent thickness of the entrance window; readout: Unidos Webline dosimeter, both PTW), positioned at the sample location, central in the SOBP.

For the irradiation of tissue slice cultures at the horizontal proton beam, a dedicated setup was built, allowing for the reproducible positioning of 6-well plates under a certain angle, relative to the proton beam. A prototype was made of Lego^®^ bricks (LEGO System A/S, Billund, Denmark), which are made of low-Z plastic material (acrylonitrile butadiene styrene), and allow for fast and easy modifications. The angle for tissue slice irradiation was optimized by means of radiochromic EBT3 film (Gafchromic, Bridgewater, MI, USA) measurements. For this purpose, EBT3 films were attached to the bottom of a 6-well plate, or inside the membrane inserts, applying different medium volumes; and irradiated under varying angles between 42° and 90° in horizontal (i.e., standing on the long side) or vertical (i.e., standing on the short side) plate direction. Two days after irradiation, the films were scanned with an Epson Expression 11,000 XL flatbed scanner (Epson, Meerbusch, Germany), and film darkening was converted to dose using the proton calibration curve implemented in a software based on Python 3.7 (Python Software Foundation, Wilmington, DC, USA).

Except for control sample data (pooled from all treatment groups) indicated sample sizes n are always technical replicates. An overview of the different treatment times and conditions is presented in Table 2. All doses were delivered in a single fraction.

### 4.5. Cytotoxicity, Viability, Inflammation

Cytotoxicity was measured via relative absorbance of lactate dehydrogenase (LDH) in the medium. The LDH assay was performed with a Pierce LDH Cytotoxicity Assay Kit (88953, Thermo Fisher Scientific, Waltham, MA, USA), according to the manufacturer’s instructions. Absorbance was measured at 490 nm and 680 nm with an Epoch^TM^ Microplate Spectrophotometer (BioTek, Bad Friedrichshall, Germany). On the last day n, remaining LDH in the tissue was released by a 4 h incubation with 1× LDH Lysis Buffer.

Cell death was calculated as followed:(1)$$LDH_{total}=LDH_{lysis}+\overset{n}{\underset{i=1}{\sum }}LDH\;of\;day(i),$$
(2)$$Cell\;death\;on\;day\left(i\right)=\frac{LDH\;of\;day\left(i\right)}{LDH_{total}},$$
(3)$$Fraction\;of\;surviving\;cells\;on\;day\left(i\right)=1-\overset{n}{\underset{i=1}{\sum }}Cell\;death\;on\;day\left(i\right).$$

Viability of the tumor was assessed via slice metabolism of PrestoBlue^TM^ (A13262, Thermo Fisher Scientific, Waltham, MA, USA) in the supernatant, according to the manufacturer’s instruction. The absorbance A was measured at 571 nm and 601 nm at the spectrophotometer. Metabolism was calculated as followed:
(4)$$A_{background}=A_{medium\;only}\left(571\;nm\right)-A_{medium\;only}\left(601\;nm\right),$$
(5)$$A_{corrected}=A_{sample}\left(571\;nm\right)-A_{sample}\left(601\;nm\right)-A_{background}.$$

For inflammation, culture medium was analyzed via either LEGEND MAX TM mouse-IL-6 Elisa-Kit (BLD-431307, Biozol Diagnostica, Eching, Germany) or LEGENDplex™ Mouse Inflammation Panel (740150, BioLegend, San Diego, CA, USA), according to the manufacturer’s instructions. Elisa plates were measured at a Tecan Plate reader, the cytometric bead array was analyzed at a BD LSR Fortessa™ (BD Biosciences, Franklin Lakes, NJ, USA).

Due to variation of the slice sizes, all data was normalized to day 1 in culture.

### 4.6. Histology

For histological analysis, the slices were harvested 24 h post irradiation, fixed in 4% formalin overnight at room temperature (RT), and paraffin embedded. To ensure flat embedding, the TSC and OBSC were wrapped in foam foil (TAP, Braunschweig, Germany) and gently pressed on with a stamp (Medite, Burgdorf, Germany) directly before embedding. All stainings were performed on 3 µm thick, dewaxed, and rehydrated sections; all washing steps were done in PBS.

H&E staining was performed according to standard procedures. For immunohistochemistry, heat induced antibody retrieval was performed with citrate buffer (pH = 6), before incubation with the respective primary antibodies against BrdU, pimonidazole, CD44, GFAP, γH2AX, Iba1, Ki67, MAP2, NeuN, and OSP. A list of antibody specifications and their corresponding secondary antibodies can be found in Appendix A.

OBSC exhibited a strong auto-fluorescence, which was quenched with 0.3% Sudan Black/EtOH counterstaining for immunofluorescence analysis. It was applied for 1 h at room temperature, followed by a 3 × 5 min washing step in PBS before primary antibody incubation.

Imaging was done at 10× or 40× magnification with an Axio Observer Z1 and an Axio Imager M1 (Carl Zeiss, Jena, Germany).

### 4.7. Data Analysis

Images were analyzed with Fiji ImageJ (Version 1.52p) [70]. Scripts for evaluation of H&E (github.com/jo-mueller/NecroticSegmentation), DAPI (github.com/Theresa-S/IF_quantification), and CD44 (github.com/Theresa-S/IHC_quantification) images (10 × magnification) are provided on GitHub; a brief description of the workflows is shown in Appendix A. A semi-automated analysis pipeline for γH2AX foci evaluation (imaged with 40 × magnification) was established using Fiji (github.com/jo-mueller/FociCounter). Automated thresholding and cell segmentation of cell nuclei, based on the DAPI-staining, was combined with a subsequent manual selection step, in order to exclude mitotic and apoptotic cells. Only visually verified, properly segmented cells with an area above 65 µm^2^ were included in the analysis. A Gaussian blur was applied, and foci were detected by counting the number of local intensity maxima. The algorithm’s results were compared with the manually determined results from two independent observers (TR, TS). Calculation of the cfoci was done, as previously described [49]. All scripts were written in ImageJ macro language.

Data handling and analysis was performed with GraphPad PRISM (Version 6.03 (2013), GraphPad Software, Inc., San Diego, CA, USA), RStudio (Version 3.5.3 (2019), RStudio Inc., Boston, MA, USA) and SPSS (Version 25, IBM, Armonk, NY, USA).

## 5. Conclusions

Taken together, adult OBSC seem to be less suitable for quantification of normal tissue reaction after irradiation, and alternatives such as neonatal brain slices or brain organoids should be considered. TSC, on the other hand, proved to be a useful tool for in situ analysis of different biomarkers such as γH2AX foci as a predictor of radiosensitivity and CD44 as a CSC marker. In an explorative study, thin-cut tissue slices of patient-derived xenografts or primary tumor materials should be evaluated for the potential to predict treatment response to various therapies, in comparison to the parental tumor’s response. In the future, this method might be applied as a fast and easy routine method to predict therapy outcome in personalized oncology.

## Figures and Tables

**Figure 1 cancers-12-01589-f001:**
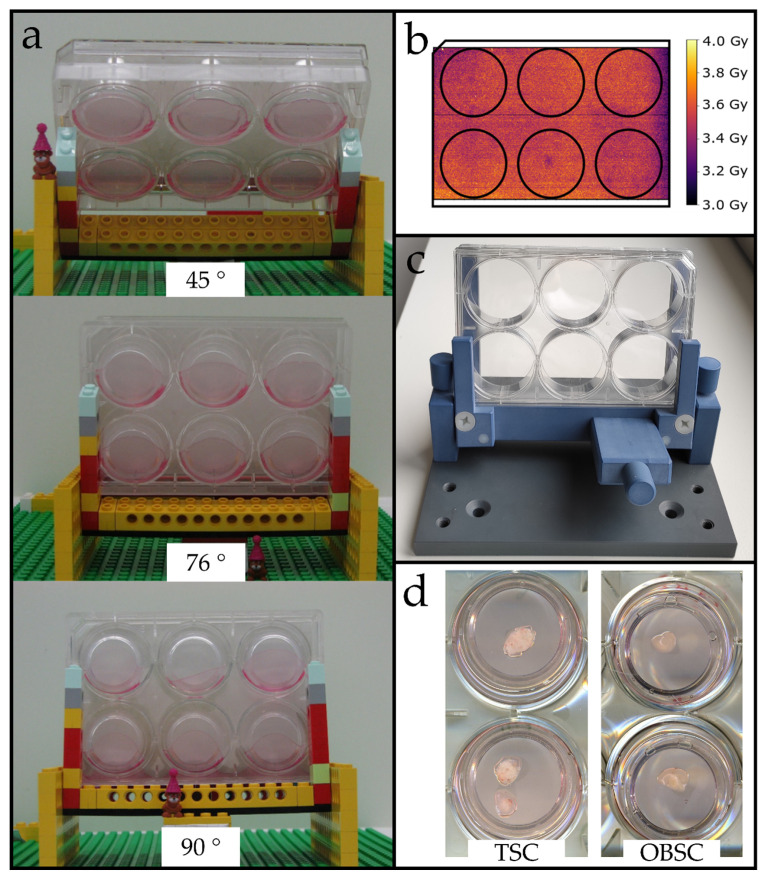
Development and characterization of a proton irradiation setup for thin-cut tissue slices. (**a**) Exemplary tested angles using a Lego^®^ rapid prototype. Membrane inserts tilted when angles >76° were used. Lower angels showed dose inhomogeneity. (**b**) Dose homogeneity for a 76° angle measured with EBT3 films on the plate bottom. (**c**) Adjustable milled setup at 76°. (**d**) Exemplary tumor (**left**) and brain slices (**right**) on the first day in culture.

**Figure 2 cancers-12-01589-f002:**
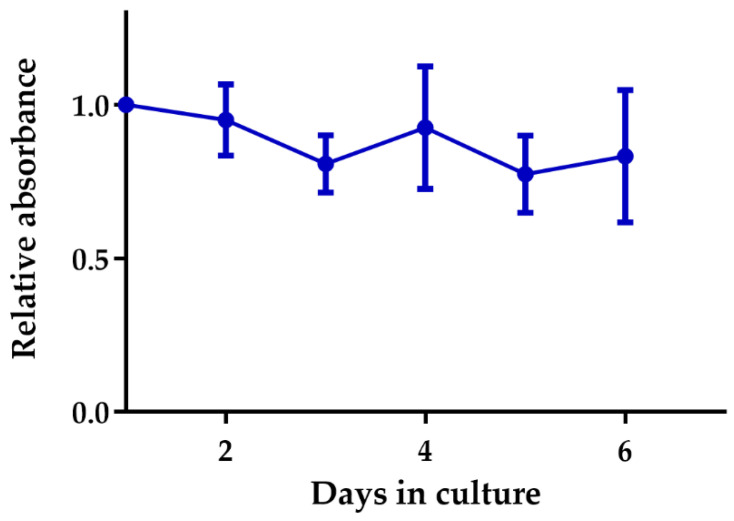
Metabolism of Cal33 tumor slices measured via PrestoBlue^TM^. Values were normalized to the first measurement at day 1 in culture. Cell viability stayed stable over the observed cultivation period of six days (mean with standard deviation (SD), n = 7).

**Figure 3 cancers-12-01589-f003:**
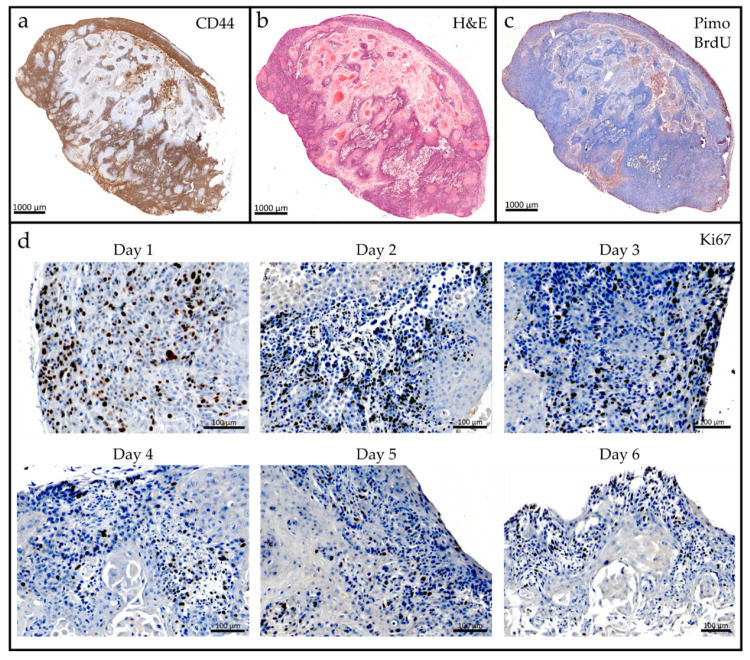
Histological analysis of tumor slice morphology and cell composition of a Cal33 squamous cell carcinoma of the head and neck (HNSCC) model. After two days in culture, tumor tissue contains (**a**) CD44 positive cells (brown), as well as (**b**) a necrotic fraction (light pink) located at the core of the tissue. (**c**) The center is hypoxic (red, pimonidazole), with proliferating cells (brown, bromodeoxyuridine (BrdU)) in the oxic rim. (**d**) Ki67 staining (brown) reveals cell proliferation across a culture period of six days. Counter staining with hematoxylin, blue cell nuclei.

**Figure 4 cancers-12-01589-f004:**
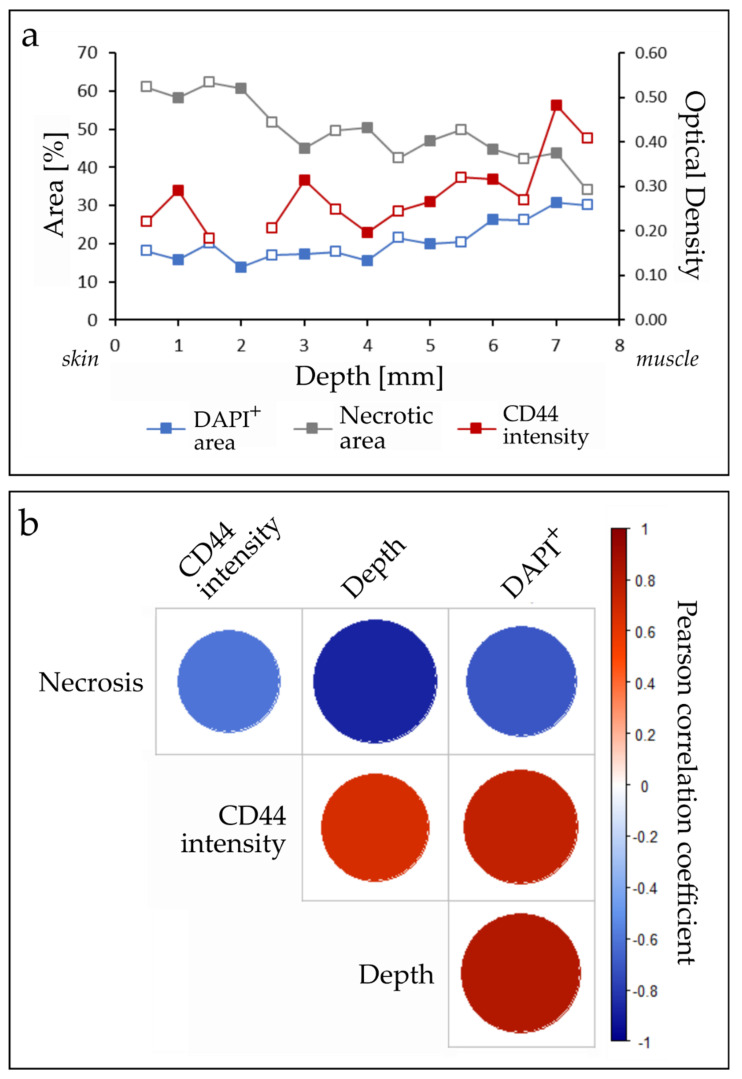
Cell composition of tumor slices along the depth of the tissue. Cancer stem cells (CD44^+^), cell numbers (DAPI^+^), and the necrotic area were analyzed 24 h post irradiation with 4 Gy or control slices. Across the tumor a high variance was noted, showing that the morphology depends on the depth within the tissue. (**a**) Marker distribution across the tumor depth: locations further away from the nutrient-providing blood vessels in the muscle show a reduced cell number and an increased necrotic area. An inter-slice variability was noted, but no significant difference between the experimental conditions (full dots: control slices, open dots: irradiated slices). (**b**) Pearson correlation of tumor morphology. Significant positive and negative correlations were found between all markers (see text, n_Control_ = 7, n_Tumor_ = 8).

**Figure 5 cancers-12-01589-f005:**
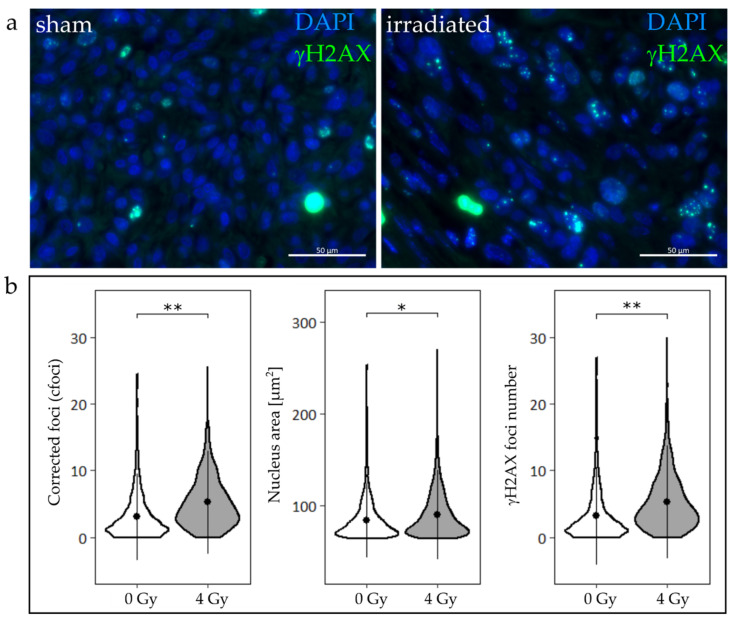
Formation of γH2AX foci in tumor slices exposed to sham (0 Gy) or 4 Gy proton irradiation at 24 h post irradiation. (**a**) Representative immunofluorescent images of tumor slices treated with sham (**left**) or proton irradiation (**right**) show γH2AX foci. Apoptotic cells and endogenous DNA damages were observed in both groups; nevertheless, an increased number of foci could be detected in irradiated slices. (**b**) cfoci, nucleus area, and γH2AX foci numbers are significantly increased in slices that were irradiated with 4 Gy, compared to non-irradiated controls (linear mixed-effects model, *: *p* < 0.05, **: *p* < 0.01; n_Control_ = 7, n_Tumor_ = 8).

**Figure 6 cancers-12-01589-f006:**
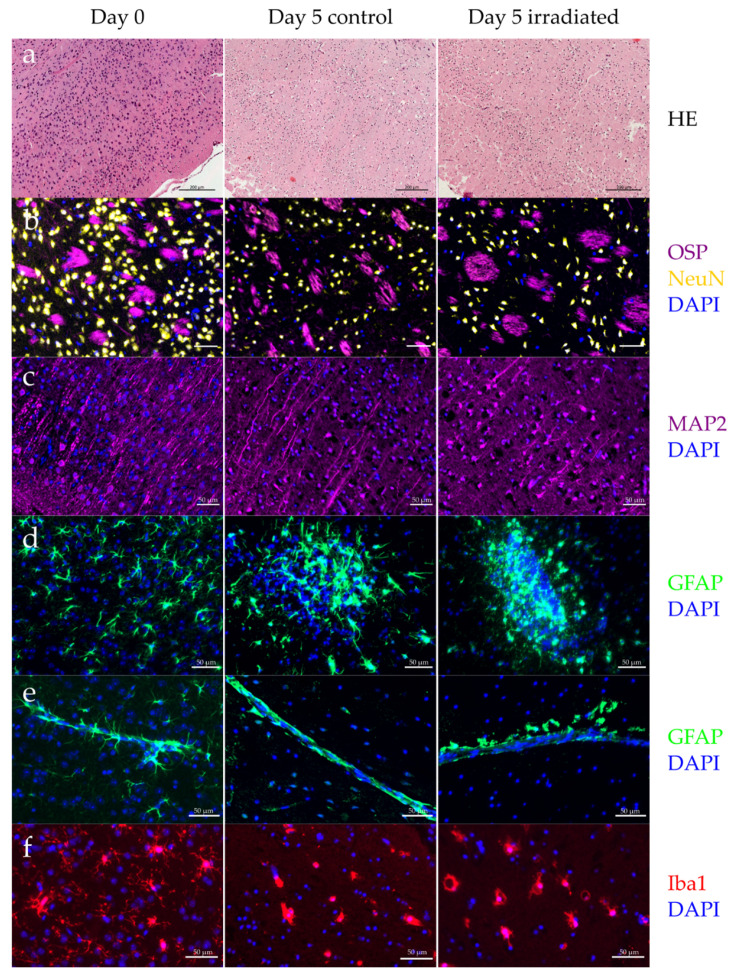
Cell type composition of organotypic brain slice culture after cutting, and at day 5 in culture for irradiated (10 Gy) and non-irradiated slices (24 h post irradiation). Both (**a**) H&E and (**b**) staining of neuronal nuclei (NeuN) revealed a shrunken cell morphology. Distribution of neurons (NeuN) and myelin (OSP) was unchanged, but (**c**) dendrite density (MAP2) decreased during cultivation. (**d**) Some astrocytes (GFAP) started uncontrolled proliferation, creating cell clusters. (**e**), (**f**) Loss of function was observed for astrocytes lining vessels and microglia (Iba1) in the tissue.

**Figure 7 cancers-12-01589-f007:**
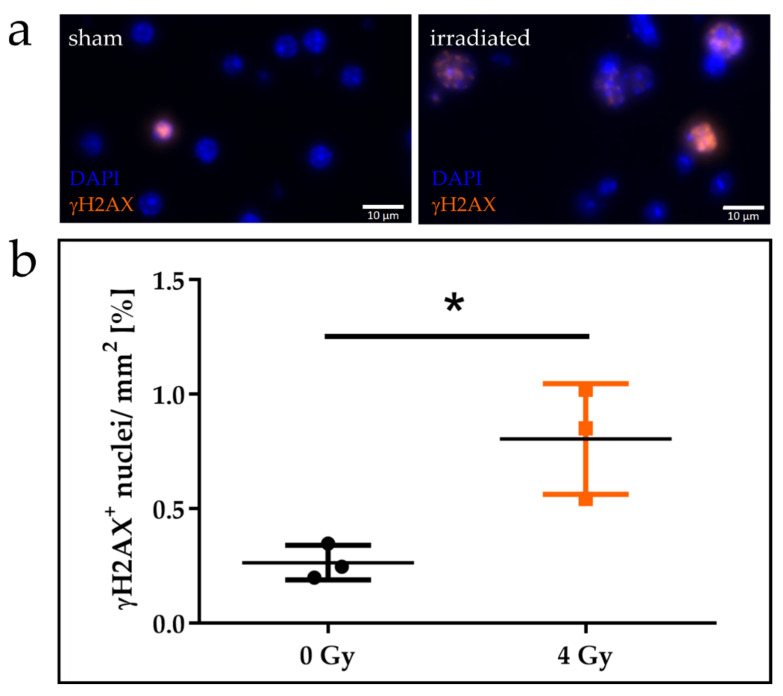
Formation of γH2AX signal in organotypic brain slice cultures irradiated with 10 Gy protons. (**a**) Irradiated slices show a higher number of γH2AX positive cells. However, the typical spot-like foci formation is missing (blue—cell nuclei, orange—γH2AX). (**b**) The percentage of γH2AX positive nuclei (mean with SD) in non-irradiated and proton irradiated organotypic brain slices revealed a significant increase after proton irradiation (unpaired, two-tailed *t*-test, n = 3, *: *p* = 0.02).

**Table 1 cancers-12-01589-t001:** Cell Culture Media-Reagents and their Respective Concentrations.

Medium	Reagent	Supplier	Cat.No.
TSC: DMEM complete	Dulbecco’s modified Eagle Medium	Thermo Fisher Scientific ^1^	61965026
	10% Fetal Bovine Serum	Sigma-Aldrich ^2^	F7524
	2% HEPES Buffer solution	Biochrom ^3^	L1613
	1% non-essential amino acids	Biochrom ^3^	K 0293
	1% sodium pyruvate	VWR ^4^	L0473
	1% Penicillin/Streptomycin	Biochrom ^3^	A2213
OBSC: Dissection buffer	Hibernate A	Thermo Fisher Scientific ^1^	A1247501
	1× Glutamax	Thermo Fisher Scientific ^1^	35050061
	1× Antibiotic-Antimycotic	Thermo Fisher Scientific ^1^	15240062
	1× B27	Thermo Fisher Scientific ^1^	17504044
OBSC: Cultivation medium	Neurobasal A	Thermo Fisher Scientific ^1^	10888022
	1× Glutamax	Thermo Fisher Scientific ^1^	35050061
	1× Antibiotic-Antimycotic	Thermo Fisher Scientific ^1^	15240062
	1× B27	Thermo Fisher Scientific ^1^	17504044

^1^ Thermo Fisher Scientific, Waltham, MA, USA; ^2^ Sigma-Aldrich, Missouri, MI, USA; ^3^ Biochrom, Berlin, Germany; ^4^ VWR, Radnor, PA, USA.

**Table 2 cancers-12-01589-t002:** Overview of the Individual Treatment Conditions and Times.

	Endpoint	Dose [Gy]	Time in Culture Before:		Animal Number	Slice Number (Treated/Control)
			Treatment	Harvest		
TSC	Histology	0, 4	24 h	48 h	1	8/7
	LDH Assay	0, 10, 20	4 d	13 d	1	3/3
OBSC	Histology	0, 10	4 d	5 d	1	3/3
	LDH Assay	0, 10–35	4 d	11 d	5	6/30

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
