# Peer review of "Applying Tissue Slice Culture in Cancer Research—Insights from Preclinical Proton Radiotherapy"

_cancers, 2020, doi:10.3390/cancers12061589_

Round 1

Reviewer 1 Report

This an interesting and well characterized effort to establish a tissue slice overlay system of HNSCC and adult OBSC and to explore the effects of proton therapy within these models.  

The main area of concern, however, is that while it is claimed that HNSCC TSC's are useful tool for in situ analysis of different radiobiologic endpoints, as the authors state there is very little evidence to suggest significant changes in several "biomarkers" associated with radiation - histoarchitecture, ncerosis, cancer stem cells, cell death, etc. So this leads to questions about technical issues and optimal time points as well as utility of the TSC system in general. To address this the authors may consider performing similar experiments in the PDOX model to see how this correlates. 

Minor comments:

It would be helpful to have some description within the Results or a citation in the Intro of the HNSCC xenografts used in study.

"Irradiation" in the Result section looking at DAPI, necrosis, and CD44 needs to be briefly explained - 1 dose, multiple, dose/fx? As well for OBSC - as a single fraction?

Reviewer 2 Report

In this manuscript, Suckert et al, investigated the relevance of thin-cut tumor slices cultures (TSC) of head and neck cancer and organotypic brain slices cultures (OBSC) to address questions pertaining to proton therapy.

The authors engineered an angular system to irradiate 6-well plates with a horizontal proton beam and measured different parameters of radiation responses in TSC and OBSC.

General comments:

  • Before performing any experiments, authors have performed dosimetry studies. Pieces of EBT3 film were placed in the bottom of the plate to ensure the homogeneity of the delivered dose. However, volume of medium used for TSC and OBSC should be added on top of the film to mimic the exact condition of proton irradiation.
  • It is not clear know many Head and Neck tumors were used to performed TSC and how many TSC were used per experiment. Please edit the manuscript accordingly.
  • Similar experiments using another tumor type should be used to confirm the validity of TSC.
  • Authors concluded that OBSC were not an appropriate model to address questions pertaining to proton irradiation and emphasize on the use of TSB. Even though parameters to compare both methods are not identical, authors should further validate their claim by comparing OBSC with thin-cut tissue slice from neonate brains.
  • Another method to evaluate cell death in TSC versus OBSC should be added.

Reviewer 3 Report

The authors work with evaluation of thin-cut (500 micrometer) tumor slice cultures of a HNSCC xenograft (Origin: Cal33 head neck cancer cell line) and organotypic slice cultures of adult mouse brain with proton beam radiation therapy, which is quite rare in research field. Endpoints (up to 14 d) with cellular survival, metabolic activity, and morphology of different cell populations were analyzed following proton beam irradiation (with Tissue Slice angle 76 degree). The DNA double strand break marker γH2AX was used to quantify the radiation response.

Tumor graft possess some characteristics while surviving in the xenograft mice model. The tumor graft over the hind leg of nude mice : Locations farer away from the nutrient-providing blood vessels in the mice muscle show a reduced cell number and an increased necrotic area, while tumor cell closing to muscle increased with CD44 ( cancer stem cells) and DAPI increased. Also, Tumor slice culture (TSC) may show advantages over primary tumor cell culture.     Once removed from mice, TSC can be cultured with cell viability for 6 days at least.

It may be useful tool for in situ analysis of biomarkers such as γH2AX as a predictor of radiosensitivity and CD44 as a CSC marker. γH2AX foci assay can be applied in TSC with semi-automated foci counting algorithm. In the Future, thin-cut tissue slices of patient-derived xenografts may be used to predict treatment outcome.

Advantage of this study: proton Radiotherapy, PBRT, is precise and the research about tumor response and PBRT is only few.   Tissue slice culture with preserve the original 3-dimensional structure and mimicking the in vivo situation rather than the primary culture may offer an easier way to propagate, thus making personalized medicine less expensive.

LIMITATIONS:

1.  HN cancer cell line, grows in nude mice, thin-sliced, culture and check viability, CSC marker or radiosensitivity. The cancer cell line own its intrinsic pre-existing radiosensitivity (or radioresistance) and cancer stem cell stemness. It can be improved by using primary tumor instead . It will be with more tumor-heterogeneity.

2.  24 hr for observation of change after radiation is relatively short even with single large dose radiation.

Minor issue :

Introduction (line 60) : esophagus is not head neck cancer.

Round 2

Reviewer 1 Report

The authors have addressed this Reviewer's concerns and have reworked the manuscript nicely. Issues related to COVID 19 understandably preclude fully addressing several of the issues raised by the Reviewers.